# The applicability of commonly used predictive scoring systems in Indigenous Australians with sepsis: An observational study

Josh Hanson[1,2]*, Simon Smith[1], James Brooks[3,4], Taissa Groch[5], Sayonne Sivalingam[3], Venessa Curnow[6], Angus Carter[3], Satyen Hargovan[3]

**1** Department of Medicine, Cairns Hospital, Cairns, Queensland, Australia, **2** The Kirby Institute, University of New South Wales, Sydney, New South Wales, Australia, **3** Department of Intensive Care Medicine, Cairns Hospital, Cairns, Queensland, Australia, **4** Department of Anaesthetics, Gloucestershire Royal Hospital, Gloucester, United Kingdom, **5** Department of Anaesthetics, Cairns Hospital, Cairns, Queensland, Australia, **6** Aboriginal and Torres Strait Islander Health, Torres and Cape Hospital and Health Service, Cooktown, Queensland, Australia

* jhanson@kirby.unsw.edu.au

**Data Availability Statement:** All relevant data are within the manuscript and its Supporting Information files.

## Abstract

### Background

Indigenous Australians suffer a disproportionate burden of sepsis, however, the performance of scoring systems that predict mortality in Indigenous patients with critical illness is incompletely defined.

### Materials and methods

The study was performed at an Australian tertiary-referral hospital between January 2014 and June 2017, and enrolled consecutive Indigenous and non-Indigenous adults admitted to ICU with sepsis. The ability of the ANZROD, APACHE-II, APACHE-III, SAPS-II, SOFA and qSOFA scores to predict death before ICU discharge in the two populations was compared.

### Results

There were 442 individuals enrolled in the study, 145 (33%) identified as Indigenous. Indigenous patients were younger than non-Indigenous patients (median (interquartile range (IQR) 53 (43–60) versus 65 (52–73) years, p = 0.0001) and comorbidity was more common (118/145 (81%) versus 204/297 (69%), p = 0.005). Comorbidities that were more common in the Indigenous patients included diabetes mellitus (84/145 (58%) versus 67/297 (23%), p<0.0001), renal disease (56/145 (39%) versus 29/297 (10%), p<0.0001) and cardiovascular disease (58/145 (40%) versus 83/297 (28%), p = 0.01). The use of supportive care (including vasopressors, mechanical ventilation and renal replacement therapy) was similar in Indigenous and non-Indigenous patients, and the two populations had an overall case-fatality rate that was comparable (17/145 (12%) and 38/297 (13%) (p = 0.75)), although Indigenous patients died at a younger age (median (IQR): 54 (50–60) versus 70 (61–76)

**Funding:** The authors received no specific funding for this work.

**Competing interests:** The authors have declared that no competing interests exist.

years, p = 0.0001). There was no significant difference in the ability of any the scores to predict mortality in the two populations.

## Conclusions

Although the crude case-fatality rates of Indigenous and non-Indigenous Australians admitted to ICU with sepsis is comparable, Indigenous patients die at a much younger age. Despite this, the ability of commonly used scoring systems to predict outcome in Indigenous Australians is similar to that of non-Indigenous Australians, supporting their use in ICUs with a significant Indigenous patient population and in clinical trials that enrol Indigenous Australians.

## Introduction

Globally, sepsis is estimated to kill 11 million people every year, predominantly in low- and middle income countries [1]. Even in the well-resourced Australian health system, sepsis kills over 5000 people annually [2]. However, the case-fatality rate is not uniform across Australia [3]. This can be partly explained by differences in the complexity of patient care at different sites, but other factors also contribute and include patient age and demographics, the geographical location of the Intensive Care Unit (ICU), and the local prevalence of different pathogens [3].

In an effort to ensure that all Australians admitted to ICU are receiving high-quality care, predictive scoring systems are used to measure patients' disease severity and determine their expected outcomes [4–8]. While these scores have limited utility in the management of individual patients, they can be used by institutions to benchmark ICU performance. They are also important in clinical trials where they can be used to help evaluate interventions by providing a measure of study patients' disease severity. However, these scoring systems are most frequently derived and validated in metropolitan referral centres in high-income settings, their predictive ability may differ in other patient populations [9, 10].

Aboriginal and Torres Strait Islander people (hereafter respectfully referred to as Indigenous Australians) are disproportionately represented in Australian ICUs and sepsis is one of the most common indications for admission [11]. Several studies have examined the characteristics of undifferentiated cohorts of Indigenous Australians admitted to ICU, and have identified that they are younger, have greater comorbidity and live more frequently in remote locations [11–14]. While, the overall case-fatality rate of Indigenous Australians is similar to that of non-Indigenous Australians in these series, there has been limited examination of the comparative performance of predictive scoring systems in Indigenous patients [11–14]. This is important, because in some parts of the country, Indigenous Australians represent a significant proportion of ICU admissions [12, 13].

This study was performed to determine the applicability of commonly used predictive scoring systems in Indigenous Australians admitted to ICU with sepsis. It was hoped that the study might validate the use of these scoring systems in both Aboriginal and Torres Strait Islander Australians, which would support their use in future clinical sepsis trials enrolling Indigenous patients. It would also provide justification for their use in benchmarking the performance of ICUs with a greater proportion of Indigenous patients [15].

## Materials and methods

This retrospective study used data collected in the ICU of Cairns Hospital, the only ICU in the Far North Queensland (FNQ) region. FNQ's population of 279,354—approximately 17% of whom identify as Indigenous Australians—is dispersed across an area of 204,255 km$^2$ [16].

Consecutive adults ($\geq$ 18 years) admitted between 1 January 2014 and 30 June 2017 to the ICU with a primary admission diagnosis of sepsis (Acute Physiology and Chronic Health Evaluation (APACHE) III-J diagnostic codes 501–504: non-urinary sepsis, urinary sepsis, non-urinary sepsis with shock, and urinary sepsis with shock respectively) were eligible for the study. Demographic, clinical and laboratory data were collected and correlated with the patients' clinical course. Comorbidities were said to be present if they were pre-existing and documented in the medical record (MetaVision$^®$) These included a history of cardiovascular disease, respiratory disease, renal disease, haematological disease or malignancy, liver disease immunocompromise or diabetes mellitus. Hazardous alcohol consumption was defined as regular consumption of greater than 4 standard drinks/day [17]. The data were collected by 4 members of the research team (SH SS TG JB) who met regularly to confirm uniformity of documentation. If patients were admitted to ICU more than once during the study period, only their first admission was included in the analysis.

All individuals receiving care in Queensland's public health system, are asked whether they identify as an Aboriginal Australian, a Torres Strait Islander Australian, both or neither. Commonly used disease severity prediction scores including the Australia and New Zealand Risk Of Death (ANZROD) score [4], the APACHE-II and APACHE-III scores [18], the Simplified Acute Physiology Score (SAPS) II score [19] and the quick Sequential Organ Failure Assessment (qSOFA) score [20] were calculated using variables collected at the time of ICU admission. The Sequential Organ Failure Assessment (SOFA) score was calculated using the worst values recorded in the first 24 hours of the ICU admission [8]. The scores' ability to predict both death prior to ICU discharge and at 90 days was determined using the patient's medical record and the Hospital Based Corporate Information System (HBCIS).

## Statistical analysis

Data were de-identified, entered in an electronic database (S1 Dataset) and analysed using statistical software (Stata version 14.2). Groups were compared using the Kruskal-Wallis, Chi-square test or Fisher's exact test where appropriate. Multivariate analysis was performed using backwards stepwise logistic regression. The Indigenous population was further examined by dividing the Indigenous population into those who identified as Aboriginal Australians and those who identified as Torres Strait Islander Australians. Those who identified as both, were not included in analyses comparing the two Indigenous populations.

The ability of the scores to predict death were determined by measuring the area under receiver operator characteristic (AUROC) curve [21]; the optimal cut-off for the tests for the different populations were determined using Liu's method [22].

## Ethics approval

Ethics approval was obtained from the Far North Queensland Human Research Ethics Committee (HREC/17/QCH/93/AMO2). As the data were retrospective and de-identified, the Committee waived the requirement for informed consent.

## Results

There were 442 individuals admitted to ICU for sepsis, on 500 occasions, during the study period. Their median (interquartile range (IQR)) age at their first presentation was 59 (48–70) years, 238 (54%) were male. Of the 442 patients, 145 (33%) identified as Indigenous Australians, 94 (65%) identified as Aboriginal, 36 (25%) identified as Torres Strait Islanders, while 15 (10%) identified as both. Among the 442 patients, 416 (94%) were FNQ residents; Indigenous patients were more likely to reside in a rural or remote location than non-Indigenous patients (88/144 (61%) versus 128/272 (47%), p = 0.006).

### Patient characteristics

Indigenous patients were younger than non-Indigenous patients (median (IQR): 53 (43–60) versus 65 (52–73) years, p = 0.0001) and were more likely to have a significant comorbidity (118/145 (81%) versus 204/297 (69%), p = 0.005).

Indigenous patients were more likely to have diabetes mellitus than non-Indigenous patients (odds ratio (OR): 4.7, 95% confidence interval (CI): 3.1–7.2), which contributed to a significantly greater burden of cardiovascular disease and renal disease (Table 1). Diabetes was more common in Torres Strait Islanders than Aboriginal Australians (27/36 (75%) versus 48/94 (51%), p = 0.01). No fewer than 32/145 (22%) Indigenous patients had a history of sepsis (hospitalisation with APACHE III-J diagnostic codes 501–504 prior to January 2014).

**Table 1. Demographic characteristics and comorbidities of the cohort, stratified by Indigenous status.**

| Variable | Indigenous n = 145 | Non-Indigenous n = 297 | p |
|---|---|---|---|
| Age | 53 (43–60) | 65 (52–73) | 0.0001 |
| Male gender | 68 (47%) | 170 (57%) | 0.04 |
| Residence in a remote location [a] | 88/144 (61%) | 128/272 (47%) | 0.006 |
| Inter-hospital transfer | 56 (39%) | 115 (39%) | 0.98 |
| Admitted from Emergency Department | 44 (30%) | 95 (32%) | 0.73 |
| Admitted from a nursing home | 0 | 0 | - |
| Planned admission after surgery | 3 (2%) | 2 (1%) | 0.34 |
| Admitted from hospital ward | 42 (29%) | 83 (28%) | 0.82 |
| Admission at night | 94 (65%) | 182 (61%) | 0.47 |
| Hazardous alcohol consumption | 58 (40%) | 53 (18%) | <0.0001 |
| Cigarette smoker | 88 (61%) | 134 (45%) | 0.002 |
| History of sepsis prior to the study period | 32 (22%) | 30 (10%) | 0.001 |
| History of cardiovascular disease | 58 (40%) | 83 (28%) | 0.01 |
| History of respiratory disease | 31 (21%) | 49 (17%) | 0.21 |
| History of renal disease | 56 (39%) | 29 (10%) | <0.0001 |
| History of haematological disease or malignancy | 7 (5%) | 29 (10%) | 0.10 |
| History of liver disease | 18 (12%) | 22 (7%) | 0.09 |
| History of diabetes mellitus | 84 (58%) | 67 (23%) | <0.0001 |
| History of metastatic cancer | 6 (4%) | 29 (10%) | 0.04 |
| Immunocompromised | 9 (6%) | 52 (18%) | 0.001 |
| Significant comorbidity | 118 (81%) | 204 (69%) | 0.005 |
| Body mass index | 27 (22–33) | 28 (24–33) | 0.07 |

Absolute numbers (%) or median (interquartile range) are presented.

[a] Only includes the 416 Far North Queensland residents.

While Indigenous patients, as a group, were more likely to have a history of hazardous alcohol or tobacco use, hazardous alcohol use was more common in the Aboriginal patients than in Torres Strait Islanders (49/94 (52%) versus 7/36 (9%) (p = 0.001). The difference in smoking rates between Aboriginal and Torres Strait Islanders failed to reach statistical significance (63/94 (67%) versus 18/36 (50%), p = 0.07). Other demographic characteristics of the cohort and their comorbidities—stratified by Indigenous status—are presented in Table 1.

## Source of sepsis and microbiological characteristics

The commonest presumed source of sepsis was the respiratory tract, although skin and soft tissue infections (SSTI) were also common and occurred more frequently in Indigenous patients, particularly those with diabetes (31/41 (76%) Indigenous patients with a SSTI were diabetic, compared with 53/104 (51%) with another source, p = 0.007). *Staphylococcus aureus* was the most commonly isolated pathogen in the cohort and was responsible for 66 (15%) admissions; 13 (18%) of the isolates were methicillin resistant. In general, however, drug resistant pathogens and "tropical" pathogens were relatively uncommon (Table 2).

**Table 2. Source and aetiology of the sepsis, stratified by Indigenous status.**

| Variable | Indigenous n = 145 | Non- Indigenous n = 297 | p |
|---|---|---|---|
| Respiratory source | 51 (35%) | 97 (33%) | 0.60 |
| Genitourinary source | 32 (22%) | 64 (22%) | 0.90 |
| Bone/joint source | 7 (5%) | 6 (2%) | 0.13 |
| Central nervous system source | 0 (0%) | 6 (2%) | 0.18 |
| Skin/soft tissue source | 41 (28%) | 43 (14%) | 0.0001 |
| Abdominal source | 13 (9%) | 43 (14%) | 0.10 |
| Other source | 10 (7%) | 47 (16%) | 0.01 |
| Bacterial infection | 103(71%) | 194 (65%) | 0.23 |
| Gram negative bacteria | 63 (43%) | 131 (44%) | 0.90 |
| Gram positive bacteria | 55 (38%) | 84 (28%) | 0.04 |
| Fungal infection | 7 (5%) | 9 (3%) | 0.42 |
| Viral infection | 10 (7%) | 22 (7%) | 0.85 |
| Drug resistant organism | 5 (3%) | 11 (4%) | 1.0 |
| Bacteraemia | 56 (39%) | 128 (43%) | 0.37 |
| Polymicrobial infection | 34 (23%) | 46 (15%) | 0.04 |
| *Escherichia coli* | 15 (10%) | 44 (15%) | 0.20 |
| *Staphylococcus aureus* | 26 (18%) | 40 (13%) | 0.22 |
| Methicillin-resistant *S. aureus* | 7 (5%) | 6 (2%) | 0.13 |
| *Pseudomonas aeruginosa* | 15 (10%) | 28 (9%) | 0.76 |
| *Klebsiella pneumoniae* | 9 (6%) | 15 (5%) | 0.66 |
| Influenza A | 5 (3%) | 13 (4%) | 0.80 |
| *Burkholderia pseudomallei* | 8 (6%) | 8 (3%) | 0.17 |
| *Streptococcus pyogenes* | 16 (11%) | 9 (3%) | 0.001 |
| Leptospirosis | 0 | 11 (4%) | 0.02 |
| *Streptococcus pneumoniae* | 6 (4%) | 10 (3%) | 0.79 |
| *Pneumocystis jirovecii* | 2 (1%) | 4 (1%) | 1.0 |
| *Mycobacterium tuberculosis* | 0 | 1 (0.3%) | 1 |
| *Plasmodium falciparum* | 0 | 1 (0.3%) | 1 |
| *Rickettsia australis* | 1 (0.7%) | 0 | 0.33 |
| *Vibrio vulnificus* | 1 (0.7%) | 0 | 0.33 |

Absolute numbers (%) presented.

### Clinical and laboratory findings at presentation

Renal impairment and metabolic acidosis were more common among Indigenous patients, but other laboratory findings in the Indigenous and non-Indigenous patients were similar (Table 3).

### Intensive care support

The supportive care provided to the Indigenous and non-Indigenous patients in the ICU was similar (Table 4). While a greater proportion of Indigenous patients required renal replacement therapy (RRT), this did not reach statistical significance in this small sample (19/145 (13%) versus 27/297 (9%), p = 0.20).

### Case-fatality rate

There were 55 deaths in the cohort prior to ICU discharge: 17/145 (12%) Indigenous patients compared with 38/297 (13%) non-Indigenous patients, p = 0.75. There were 14 (15%) deaths among the 94 Aboriginal Australians and 2 (6%) among the 36 Torres Strait Islander Australians (p = 0.23). There was one (7%) death among the 15 patients who identified as both Aboriginal and Torres Strait Islander Australians.

Patients that died before ICU discharge were older than those who survived (median (IQR): 62 (54–73) versus 59 (46–69 years), p = 0.01) and more likely to have a significant comorbidity (47/55 (85%) versus 275/387 (71%), p = 0.03). Among the 416 FNQ residents in the cohort, 29/50 (58%) living in a rural or remote location died compared with 187/366 (51%), with an urban address (p = 0.36). In a multivariate analysis model which included age, Indigenous status, significant comorbidity and remote or rural residence, only age had a statistically significant association with death (OR: 1.03 (95%CI): 1.01–1.05), p = 0.007).

After 90 days, 93/442 (21%) had died. Death at 90 days was linked to age (p = 0.0001), but not to Indigenous status (p = 0.26) or rural/remote residence (p = 0.29). Indigenous patients, however, died at a younger age than non-Indigenous patients (median (IQR): 56 (52–62) versus 68 (61–76) years, p = 0.0001).

### Disease severity scores and their ability to predict death

The severity scores of the Indigenous and non-Indigenous patients were similar (Table 5). There was no statistically significant difference in the ability of the various severity scores to predict death in Indigenous and non-Indigenous patients (Tables 6 and 7). Among Indigenous patients, the ANZROD and APACHE-III scores had the highest AUROC curve (0.85 (95% CI: 0.77–0.92 and 0.84 (95% CI: 0.76–0.92), although these values were only statistically superior to that of the qSOFA scores (Tables 6 and 7). The very low case-fatality rate in the Torres Strait Islanders and relatively small sample size precluded meaningful comparison of the relative performance of the prediction scores in Aboriginal and Torres Strait Islanders.

## Discussion

In this cohort of ICU patients with sepsis there were significant differences in the age, burden of comorbidities and source of infection between Indigenous and non-Indigenous individuals. However, the ability of commonly used disease severity scoring systems to predict mortality in the two populations was similar.

FNQ, in tropical Australia, shares a border with Papua New Guinea and has a unique blend of infectious diseases; it has the country's highest incidence of leptospirosis and an increasing incidence of melioidosis and rickettsial disease [23–25]. The rates of methicillin-resistance in

**Table 3. Clinical and laboratory findings at presentation, stratified by Indigenous status.**

| Variable | Indigenous n = 145 | Non- Indigenous n = 297 | p |
|---|---|---|---|
| Heart rate (beats/min) | 99 (89–118) | 97 (81–114) | 0.28 |
| Systolic blood pressure (mmHg) | 107 (94–118) | 106 (94–122) | 0.42 |
| Mean arterial pressure (mmHg) | 73 (66–82) | 72 (65–83) | 0.50 |
| Temperature (˚C) | 36.8 (36.5–37.4) | 36.9 (36.5–37.3) | 0.53 |
| Respiratory rate (breaths/min) | 20 (17–26) | 20 (16–25) | 0.99 |
| Glasgow Coma Score | 15 (14–15) | 15 (14–15) | 0.98 |
| Glucose (mmol/L) | 6.9 (5.0–9.4) | 6.6 (5.4–8.5) | 0.96 |
| Lactate (mmol/L) | 1.8 (1.1–3.0) | 1.7 (1.1–2.8) | 0.29 |
| pH | 7.36 (7.24–7.42) | 7.37 (7.30–7.43) | 0.13 |
| $PaO_2/FiO_2$ | 291 (194–398) | 269 (158–375) | 0.02 |
| $PaCO_2$ (mmHg) | 34 (27–39) | 34 (29–42) | 0.0496 |
| Bicarbonate (mmol/L) | 18 (15–21) | 20 (17–23) | 0.001 |
| Base excess (mmol/L) | -6.2 (-10.9 to -2.8) | -4.5 (-8.4 to -2.2) | 0.004 |
| Anion gap | 9 (7–12) | 9 (7–12) | 0.44 |
| Haemoglobin (g/dL) | 100 (87–122) | 111 (93–124) | 0.007 |
| White cell count (x$10^9$/L) | 13.8 (9.3–21.9) | 13.8 (7.9–22.2) | 0.67 |
| Neutrophils (x$10^9$/L) | 10.7 (7.1–19.2) | 11.6 (6.2–19.4) | 0.80 |
| Eosinophils (x$10^9$/L) | 0 (0–0.1) | 0(0–0) | 0.001 |
| Platelets (x$10^9$/L) | 170 (111–257) | 160 (101–236) | 0.17 |
| C-reactive protein (mg/L)) | 165 (67–295) | 172 (85–286) | 0.57 |
| Troponin I (ng/mL) | 0.07 (0.04–0.39) | 0.11 (0.04–0.43) | 0.85 |
| Prothrombin (seconds) | 16 (14–21) | 16 (14–18) | 0.07 |
| APTT (seconds) | 39 (34–47) | 36 (31–40) | 0.0001 |
| INR | 1.5 (1.3–1.9) | 1.4 (1.3–1.6) | 0.03 |
| Fibrinogen (g/L) | 6.2 (4.6–8.3) | 6.6 (4.9–8.2) | 0.35 |
| Total Bilirubin (µmol/L) | 18 (12–31) | 20 (13–31) | 0.31 |
| Conjugated bilirubin (µmol/L) | 8 (4–19) | 7 (4–15) | 0.43 |
| Albumin (g/L) | 24 (20–27) | 25 (22–29) | 0.007 |
| Protein (g/L) | 58 (52–65) | 52 (48–60) | 0.0001 |
| AST (IU/mL) | 45 (21–104) | 52 (27–106) | 0.12 |
| ALT (IU/mL) | 25 (12–42) | 35 (20–65) | 0.0001 |
| GGT (IU/mL) | 41 (22–65) | 51 (26–94) | 0.01 |
| ALP (IU/mL) | 91 (69–134) | 82 (57–119) | 0.01 |
| LDH (IU/mL) | 333 (248–482) | 329 (239–439) | 0.43 |
| Sodium (mmol/L) | 133 (131–137) | 135 (133–138) | 0.0004 |
| Potassium (mmol/L) | 4.1 (3.6–4.8) | 4.0 (3.7–4.5) | 0.31 |
| Chloride (mmol/L) | 104 (99–109) | 104 (100–108) | 0.92 |
| Creatinine (µmol/L) | 162 (87–403) | 118 (75–190) | 0.0003 |
| eGFR (ml/min/1.73m$^2$) | 29 (11–65) | 49 (24–80) | 0.0002 |
| Calcium (mmol/L) | 2.2 (2.1–2.3) | 2.2 (2.1–2.3) | 0.99 |
| Magnesium (mmol/L) | 0.73 (0.66–0.87) | 0.75 (0.65–0.88) | 0.48 |
| Phosphate (mmol/L) | 1.5 (1.0–2.0) | 1.1 (0.9–1.5) | 0.0001 |

Median (Interquartile range) presented. $PaO_2/FiO_2$: ratio of arterial oxygen partial pressure (PaO2 in mmHg) to fractional inspired oxygen. $PaCO_2$: arterial carbon dioxide partial pressure. APTT: activated partial thromboplastin time; INR: International normalised ratio; AST: aspartate aminotransferase; ALT: alanine aminotransferase; GGT: Gamma-glutamyl transferase; ALP: alkaline phosphatase; LDH: lactate dehydrogenase; eGFR: estimated glomerular filtration rate.

**Table 4. Intensive care support delivered to the cohort, stratified by Indigenous status.**

| Variable | Indigenous n = 145 | Non- Indigenous n = 297 | p |
|---|---|---|---|
| Vasopressors on admission | 102 (70%) | 204 (69%) | 0.72 |
| Number of vasopressors required | 1 (1–1) | 1 (0–1) | 0.89 |
| Antibiotics administered on admission | 142 (98%) | 288 (97%) | 0.56 |
| Endotracheal intubation | 28 (19%) | 64 (22%) | 0.59 |
| Minute ventilation (Litres) | 7.3 (5.8–8.8) | 7.7 (6.6–9.0) | 0.30 |
| PICC line | 40 (28%) | 89 (30%) | 0.61 |
| Central venous line | 61 (42%) | 121 (41%) | 0.79 |
| Arterial line | 122 (84%) | 246 (83%) | 0.73 |
| Nasogastric feeding | 28 (19%) | 63 (21%) | 0.64 |
| Indwelling urinary catheter | 117 (81%) | 252 (85%) | 0.27 |
| Renal replacement therapy | 19 (13%) | 27 (9%) | 0.20 |

Absolute numbers (%) and median (interquartile range) presented. PICC: Peripherally inserted central catheter

**Table 5. Severity score values on admission to ICU, stratified by Indigenous status.**

| Variable | number | Indigenous n = 145 | Non- Indigenous n = 297 | p |
|---|---|---|---|---|
| qSOFA | 348 | 1 (1–2) | 1 (1–2) | 0.83 |
| SOFA | 378 | 9 (6–11) | 8 (6–11) | 0.20 |
| ANZROD | 430 | 0.24 (0.10–0.44) | 0.23 (0.10–0.46) | 0.86 |
| APACHE-II | 430 | 21 (15–26) | 20 (15–26) | 0.65 |
| APACHE-III | 430 | 70 (52–87) | 69 (53–87) | 0.87 |
| SAPS-II | 413 | 35 (25–51) | 38 (28–58) | 0.12 |

Median (interquartile range) presented. qSOFA: quick SOFA score; SOFA: Sequential Organ Failure Assessment score; ANZROD: Australia and New Zealand Risk Of Death score; APACHE-II: Acute Physiology, Age, Chronic Health Evaluation-II score; and APACHE-III: Acute Physiology, Age, Chronic Health Evaluation-III score; SAPS-II: Simplified Acute Physiology Score.

**Table 6. Performance of the severity scores in predicting death before ICU discharge, stratified by Indigenous status.**

| Severity score | Number of Indigenous patients in whom the score could be calculated | AUROC (95% CI) | Optimal cut-off in Indigenous patients | Number of Non- Indigenous patients in whom the score could be calculated | AUROC (95% CI) | Optimal cut-off in Non-Indigenous patients | p |
|---|---|---|---|---|---|---|---|
| qSOFA | 116 (80%) | 0.71 (0.57–0.84) | 2 | 232 (78%) | 0.58 (0.46–0.70) | 2 | 0.17 |
| SOFA | 120 (83%) | 0.75 (0.64–0.87) | 10 | 258 (87%) | 0.72 (0.63–0.80) | 10 | 0.62 |
| ANZROD | 138 (95%) | 0.85 (0.77–0.92) | 0.33 | 292 (98%) | 0.79 (0.71–0.87) | 0.35 | 0.32 |
| APACHE-II | 138 (95%) | 0.78 (0.64–0.91) | 30 | 292 (98%) | 0.76 (0.67–0.84) | 27 | 0.61 |
| APACHE-III | 138 (95%) | 0.84 (0.76–0.92) | 82 | 292 (98%) | 0.79 (0.71–0.87) | 88 | 0.37 |
| SAPS-II | 136 (94%) | 0.80 (0.70–0.90) | 46 | 277 (93%) | 0.85 (0.78–0.92) | 66 | 0.40 |

qSOFA: quick SOFA score; SOFA: Sequential Organ Failure Assessment score; ANZROD: Australia and New Zealand Risk Of Death score; APACHE-II: Acute Physiology, Age, Chronic Health Evaluation-II score; and APACHE-III: Acute Physiology, Age, Chronic Health Evaluation-III score; SAPS-II: Simplified Acute Physiology Score.

**Table 7. Performance of the severity scores in predicting 90-day mortality, stratified by Indigenous status.**

| Severity score | Number of Indigenous patients in whom the score could be calculated | AUROC (95% CI) | Optimal cut-off in Indigenous patients | Number of Non- Indigenous patients in whom the score could be calculated | Non-Indigenous n = 297 | Optimal cut-off in Non-Indigenous patients | p |
|---|---|---|---|---|---|---|---|
| qSOFA | 116 (80%) | 0.66 (0.53–0.79) | 2 | 232 (78%) | 0.59 (0.50–0.68) | 2 | 0.42 |
| SOFA | 120 (83%) | 0.75 (0.64–0.85) | 11 | 258 (87%) | 0.66 (0.58–0.74) | 9 | 0.22 |
| ANZROD | 138 (95%) | 0.85 (0.79–0.92) | 0.33 | 292 (98%) | 0.78 (0.71–0.84) | 0.29 | 0.09 |
| APACHE-II | 138 (95%) | 0.74 (0.62–0.85) | 30 | 292 (98%) | 0.72 (0.65–0.79) | 21 | 0.78 |
| APACHE-III | 138 (95%) | 0.85 (0.78–0.92) | 73 | 292 (98%) | 0.77 (0.71–0.84) | 75 | 0.13 |
| SAPS-II | 136 (94%) | 0.78 (0.68–0.88) | 36 | 277 (93%) | 0.75 (0.69–0.82) | 39 | 0.66 |

qSOFA: quick SOFA score; SOFA: Sequential Organ Failure Assessment score; ANZROD: Australia and New Zealand Risk Of Death score; APACHE-II: Acute Physiology, Age, Chronic Health Evaluation-II score; and APACHE-III: Acute Physiology, Age, Chronic Health Evaluation-III score; SAPS-II: Simplified Acute Physiology Score.

*S. aureus* isolates are among the highest reported in the country, while antibiotic resistance in other common pathogens is also increasing [26, 27]. The population is widely dispersed across a large geographical area and there are very limited specialist services outside the administrative hub of Cairns. It is a region which contains 3 of the 10 most socio-economically disadvantaged local government areas in the country, all 3 are communities with a predominantly Indigenous population [28]. It is the only part of Australia which has the homelands of both Aboriginal and Torres Strait Islander Australians, peoples that are frequently conflated but who are ethnologically quite distinct. All these factors might be expected to have implications for local patterns of sepsis, its presentation and its outcomes [15, 29–31]. However, while Indigenous patients were younger, more likely to live in a rural/remote location and more likely to have a significant comorbidity than non-Indigenous patients, the severity of their sepsis–as determined by commonly used predictive scoring systems was similar to that of non-Indigenous patients.

Although Indigenous Australians bear a greater burden of sepsis, there are surprisingly few studies that examine sepsis in the Indigenous population systematically. The demography and comorbidities in our series are very similar to that of a prospective study of sepsis in the Northern Territory, which also included patients that were not admitted to the ICU [15]. That study identified that Indigenous Australians were over-represented in the cohort, suffered disproportionately from diabetes and renal disease, and reported higher rates of smoking and hazardous alcohol use. Although APACHE-II and SOFA scores were collected in the study, the comparability of their performance in Indigenous and non-Indigenous populations was not presented.

While there are relatively few published data examining the ICU care of Australian Indigenous patients with sepsis specifically, there are several studies that have examined the clinical characteristics and outcomes of Indigenous patients admitted to ICU [11–14]. These studies are strikingly similar and show that, as in our cohort, Indigenous patients with critical illness are younger, have greater comorbidity and are more frequently admitted from remote locations than non-Indigenous patients. These studies also universally show that there is no difference in the proportion of Indigenous and non-Indigenous patients who die in the ICU. Whilst

some authors have suggested that Indigenous deaths in ICU are therefore "healthcare preventable" [13], this represents a narrow view of healthcare, de-emphasising the primary holistic care that should prevent the hospitalisation in the first place. Whilst other authors have noted that there is "no mortality gap" between Indigenous and non-Indigenous Australians admitted to ICU [11], this overlooks the fact the Indigenous patients who are admitted to ICU are younger and are dying at a younger age. In an ICU series from the Northern Territory, 7% of both Indigenous and non-Indigenous patients admitted to the ICU died, but the Indigenous patients who died were much younger than their non-Indigenous counterparts (mean age of 45 versus 56) [12]. This pattern was seen in our series: although the proportion of Indigenous patients and non-Indigenous patients dying before ICU discharge was similar, the median age of death of Indigenous patients (56 years) was 12 years lower than that of the non-Indigenous patients. To close this gap, we need evidence-based strategies to address the unique challenges faced by Indigenous Australians in the country's health system. Our study shows that predictive scoring systems are a valid way of measuring disease severity among Indigenous Australians with sepsis and can therefore be used to compare the efficacy of interventions with confidence.

Our study has many limitations. As a single centre study, it reflects only the experience of a unique part of Australia; the applicability of the results to the broader Indigenous patient population requires validation, although the similarity between the Indigenous population seen in this study and those in other locations is, as previously noted, striking [11–15]. The study was retrospective and examined only patients admitted to ICU and will therefore underestimate the true sepsis burden [32]. The sample size was relatively small, increasing the risk of a type II error.

Clearly there is still much to do to address the disparity in health outcomes between Indigenous and non-Indigenous Australians [31]. Once Indigenous Australians with sepsis enter ICU, they receive high quality care, but they are still dying at a much younger age than their non-Indigenous counterparts. Interventions at a community level including efforts to facilitate access to care, reduce crowding, enhance health literacy, improve sanitation and ensure appropriate nutrition are likely to be most helpful [33, 34]. At a primary health level optimising management of conditions like diabetes that predispose to sepsis and ensuring comprehensive vaccination, particularly for those at high risk are also likely to assist. Expanded programmes to assist with smoking cessation and encourage alcohol moderation are also essential [35, 36]. Community controlled health services which have a focus on prevention, early intervention and comprehensive care, may be the best suited to deliver this care [37, 38]. These services are frequently provided by members of the local community who have a greater understanding of the personal, community, and environmental factors influencing the health of the people and are therefore may be more likely to provide effective care [39].

However, even with optimal preventative interventions, patients will still require ICU care for sepsis. This study suggests that standard predictive scores predict outcomes in Indigenous patients as well as they do in non-Indigenous patients and therefore may be used in future studies to examine strategies to enhance the care of all Australians.

## Supporting information

**S1 Dataset.**
(XLSX)

## Acknowledgments

The authors would like to acknowledge all the health workers who were involved in the care of the patients.

## Author Contributions

**Conceptualization:** Josh Hanson, Simon Smith, Angus Carter, Satyen Hargovan.

**Data curation:** James Brooks, Taissa Groch, Sayonne Sivalingam, Satyen Hargovan.

**Formal analysis:** Josh Hanson.

**Investigation:** James Brooks, Taissa Groch, Sayonne Sivalingam, Angus Carter, Satyen Hargovan.

**Methodology:** Josh Hanson, Satyen Hargovan.

**Supervision:** Josh Hanson, Simon Smith, Venessa Curnow, Angus Carter.

**Visualization:** Simon Smith, Angus Carter.

**Writing – original draft:** Josh Hanson.

**Writing – review & editing:** Josh Hanson, Simon Smith, James Brooks, Taissa Groch, Sayonne Sivalingam, Venessa Curnow, Angus Carter, Satyen Hargovan.

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
