## [Decision Letter · Decision Letter 0]

9 Jun 2020

PONE-D-20-14432

The applicability of commonly used predictive scoring systems in Indigenous Australians with sepsis: an observational study

PLOS ONE

Dear Dr. Hanson,

Thank you for submitting your manuscript to PLOS ONE. After careful consideration, we feel that it has merit but does not fully meet PLOS ONE’s publication criteria as it currently stands. Therefore, we invite you to submit a revised version of the manuscript that addresses the points raised during the review process.

We look forward to receiving your revised manuscript.

Kind regards,

Biswadev Mitra, MBBS, MHSM, PhD, FACEM

Academic Editor

PLOS ONE

Journal Requirements:

Reviewers' comments:

Reviewer's Responses to Questions

**Comments to the Author**

1. Is the manuscript technically sound, and do the data support the conclusions?

Reviewer #1: Yes

Reviewer #2: Yes

Reviewer #3: Yes

2. Has the statistical analysis been performed appropriately and rigorously? 

Reviewer #1: I Don't Know

Reviewer #2: I Don't Know

Reviewer #3: I Don't Know

3. Have the authors made all data underlying the findings in their manuscript fully available?

Reviewer #1: Yes

Reviewer #2: Yes

Reviewer #3: Yes

4. Is the manuscript presented in an intelligible fashion and written in standard English?

Reviewer #1: Yes

Reviewer #2: No

Reviewer #3: Yes

5. Review Comments to the Author

Reviewer #1: Thank you for the opportunity to review this important piece of work. The single centre retrospective study presents original research, not elsewhere reported, comparing the ability of several mortality predicting scoring tools to predict mortality in ICU and at 90 days in Aboriginal and Torres Strait Islander peoples and non-Indigenous Australians, with sepsis. The authors report a similar predictive ability of the ANZROD, APACHE II, APACHE III, SAPS 2, qSOFA and SOFA scores in predicting death in Indigenous and non-Indigenous patients with sepsis over a 3-4 year period.

I have made several comments for the authors to consider in the attached PDF manuscript file and it may be easier for the authors to review this as opposed to me listing all of the notations here, but I outline major concerns below, in addition to minor concerns (highlighted with comments in the attached PDF).

1. Taking into account the NHMRC guideline on ethical conduct in research with Aboriginal and Torres Strait Islander peoples and communities I am curious as to whether the authors are familiar with this document and/or whether any Aboriginal and/or Torres Strait Islander peoples or communities were involved in any aspect of the study or in drafting of the manuscript?

The six core guiding principles of this guideline are key to any research undertaken with Aboriginal and Torres Strait Islander people and communities, however it was not clear from the manuscript whether the authors have considered these principles:

• respects the shared values of Aboriginal and Torres Strait Islander Peoples

• is relevant for Aboriginal and Torres Strait Islander priorities, needs and aspirations

• develops long-term ethical relationships among researchers, institutions and sponsors

• develops best practice ethical standards of research.

I recommend the authors consider these guidelines and incorporate a reflection of how they have considered these guidelines into the manuscript; methods, results and discussion. If there has not been any consultation with Indigenous persons regarding the research, this is a major limitation (and justification for rejection of the manuscript) and should be added to the limitations within the discussion. I suggest, if the journal has not already done so, that they consider enlisting an Aboriginal and/or Torres Strait Islander researcher to review this article and provide comment from an Aboriginal and/or Torres Strait Islander perspective. I am happy to make recommendations of appropriate reviewers, if needed.

2. The authors state that the primary objective of the study was to compare mortality predicting scoring systems in Indigenous and non-indigenous Australians. However, I feel the methods and results also emphasize understanding important characteristics of the populations and comparisons of these populations and that this should be considered.

I would be happy to review an edited version of the manuscript should the journal be able to seek peer review from an Aboriginal and/or Torres Strait Islander researcher.

Reviewer #2: Although Aboriginal and Torres Strait Islander is longer to write it is the preferred terminology for Aboriginal and Torres Strait Islander people in Australia. Please change throughout the paper. Another preferred terminology that you might like to use is Australia's First Nation people.

Line 186 page 8

Please indicate what supportive care means

Line 221 page 10

Different font colour

Line 231 page 10

“It is the only part of Australia which has the homelands of both Aboriginal and Torres Strait Islander Australians, peoples that are frequently conflated but who are ethnologically quite distinct”

Has “conflated” been recognised and confirmed by the community members? This can’t be just assumed. If this has not had input from community members, please remove word from this sentence.

“ethnologically” This word is a derogative term please remove it.

This sentence needs rewording to capture Aboriginal and Torres Strait Islander people’s distinct cultural groups.

Reviewer #3: The information provided with this article is not new to anyone familiar with the health determinants of the Australian Indigenous population in North Australia.

However, I think it is important to publish this research as it will stress/reiterate the need for more research and also the feasibility of using scoring systems for comparison of patient for multicentre research as well as country wide and internationally.

6. PLOS authors have the option to publish the peer review history of their article (what does this mean?). If published, this will include your full peer review and any attached files.

Reviewer #1: No

Reviewer #2: No

Reviewer #3: Yes: Ulrich Orda, NWHHS, James Cook University, Australia

---

## [Author Response · Author response to Decision Letter 0]

21 Jun 2020

Please see word document - "Response to reviewers" - uploaded as a part of the submission

---

## [Decision Letter · Decision Letter 1]

2 Jul 2020

PONE-D-20-14432R1

The applicability of commonly used predictive scoring systems in Indigenous Australians with sepsis: an observational study

PLOS ONE

Dear Dr. Hanson,

Thank you for submitting your manuscript to PLOS ONE. After careful consideration, we feel that it has merit but does not fully meet PLOS ONE’s publication criteria as it currently stands. Therefore, we invite you to submit a revised version of the manuscript that addresses the points raised during the review process.

We look forward to receiving your revised manuscript.

Kind regards,

Biswadev Mitra, MBBS, MHSM, PhD, FACEM

Academic Editor

PLOS ONE

Reviewers' comments:

Reviewer's Responses to Questions

**Comments to the Author**

1. If the authors have adequately addressed your comments raised in a previous round of review and you feel that this manuscript is now acceptable for publication, you may indicate that here to bypass the “Comments to the Author” section, enter your conflict of interest statement in the “Confidential to Editor” section, and submit your "Accept" recommendation.

Reviewer #2: (No Response)

Reviewer #3: (No Response)

2. Is the manuscript technically sound, and do the data support the conclusions?

Reviewer #2: Yes

Reviewer #3: Yes

3. Has the statistical analysis been performed appropriately and rigorously? 

Reviewer #2: Yes

Reviewer #3: I Don't Know

4. Have the authors made all data underlying the findings in their manuscript fully available?

Reviewer #2: Yes

Reviewer #3: Yes

5. Is the manuscript presented in an intelligible fashion and written in standard English?

Reviewer #2: Yes

Reviewer #3: Yes

6. Review Comments to the Author

Reviewer #2: This is a very important subject for Aboriginal and Torres Strait Islander people and this manuscript has covered the subject well. The manuscript is well written and should be published.

Reviewer #3: I think there is a typo in line 278.

shouldn't it read: "... and have a significantLY HIGHER comorbidity ...

Two other comments with the second read of the revised subscription (not necessarily for correction but if a revision is requested to consider:

1) The title of the manuscript focusses on predictive scoring systems.

The conclusion however discusses in its first sentence the case fatality and that the Indigenous patients die at a younger age. Whilst this is a very important fact / statement (and not new), this was not the question of this study. The conclusion in regard to the question of the study is mentioned (like a dependend variable) in the second sentence / statement. Would it be worth to swab them?

2) I still think that from 2014 - 2017 Cairns Hospital was NOT addressed as a Tertiary Hospital but as a retrieval "base" hospital. This is reflected by the CSCF (and also the ACEM accreditation for their Emergency Department) I know that there is a recent push / move towards addressing Cairns Hospital as a Tertiary facility (despite the lack of especially neurosurgery and cardiothoracic surgery)- but this was definitely not applicable at the time where the study data were collected,

7. PLOS authors have the option to publish the peer review history of their article (what does this mean?). If published, this will include your full peer review and any attached files.

Reviewer #2: No

Reviewer #3: No

---

## [Author Response · Author response to Decision Letter 1]

2 Jul 2020

The response to the reviewers is also provided as colour coded word document in the submission

Reviewers' comments:

Reviewer's Responses to Questions

Comments to the Author

1. If the authors have adequately addressed your comments raised in a previous round of review and you feel that this manuscript is now acceptable for publication, you may indicate that here to bypass the “Comments to the Author” section, enter your conflict of interest statement in the “Confidential to Editor” section, and submit your "Accept" recommendation.

Reviewer #2: (No Response)

Reviewer #3: (No Response)

2. Is the manuscript technically sound, and do the data support the conclusions?

Reviewer #2: Yes

Reviewer #3: Yes

Response: We are happy that the reviewers believe that the manuscript is technically sound. 

3. Has the statistical analysis been performed appropriately and rigorously?

Reviewer #2: Yes

Reviewer #3: I Don't Know

Response: We are happy that the reviewers have not raised any concerns about the statistical analysis. 

4. Have the authors made all data underlying the findings in their manuscript fully available?

Reviewer #2: Yes

Reviewer #3: Yes

Response: We are happy that the reviewers have had the opportunity to review all the data supporting our study. 

5. Is the manuscript presented in an intelligible fashion and written in standard English?

Reviewer #2: Yes

Reviewer #3: Yes

Response: We are happy that the reviewers have found our manuscript intelligible. 

6. Review Comments to the Author

Reviewer #2: This is a very important subject for Aboriginal and Torres Strait Islander people and this manuscript has covered the subject well. The manuscript is well written and should be published.

Response: We are happy that reviewer #2 has recommended publication of our manuscript. 

Reviewer #3: I think there is a typo in line 278.

shouldn't it read: "... and have a significantLY HIGHER comorbidity ...

Response: We agree with the reviewer that the sentence was poorly constructed and contained an additional typographic error. We have amended it in the revised manuscript (lines 277-278).

Two other comments with the second read of the revised subscription (not necessarily for correction but if a revision is requested to consider:

1) The title of the manuscript focusses on predictive scoring systems.

The conclusion however discusses in its first sentence the case fatality and that the Indigenous patients die at a younger age. Whilst this is a very important fact / statement (and not new), this was not the question of this study. The conclusion in regard to the question of the study is mentioned (like a dependend variable) in the second sentence / statement. Would it be worth to swab them?

Response: We understand the point that the reviewer is making here. The focus of the study was severity scores, so the reviewer is suggesting that severity scores be mentioned in the first line of the conclusion.

This is largely a question of style, however we would also argue that 

1. As death is the dependent variable for prediction scores, it makes sense to describe the variable in more detail, to address the possibility of confounding factors.

2. As the reviewer notes this is an important fact and while sadly not a new observation for Australian readers, it will help the International reader - who may not be familiar with this information – in their interpretation of our findings.

2) I still think that from 2014 - 2017 Cairns Hospital was NOT addressed as a Tertiary Hospital but as a retrieval "base" hospital. This is reflected by the CSCF (and also the ACEM accreditation for their Emergency Department) I know that there is a recent push / move towards addressing Cairns Hospital as a Tertiary facility (despite the lack of especially neurosurgery and cardiothoracic surgery)- but this was definitely not applicable at the time where the study data were collected

Response: We understand the point that the reviewer is making here. However we would argue that a hospital with advanced diagnostics (MRI, PET scanning, molecular laboratory) and which has over 500 beds (including adult and neonatal ICU) and which provides specialist medical, surgical, psychiatric, obstetric, paediatric and cancer care (including radiotherapy) is a tertiary referral hospital. 

It would appear to satisfy most definitions of the term.

https://en.wikipedia.org/wiki/Tertiary_referral_hospital

It is true that there is no neurosurgery or cardiothoracic surgery but many of the authors have trained in metropolitan tertiary centres in Australia which have no paediatric, obstetric or neonatal facilities and yet they would still be recognised as tertiary centres.

We think that most practicing clinicians would understand what a tertiary centre is and would agree with our assessment. We feel that this is more than a semantic argument; it is is important that the reader can understand the context of the patient’s care and the impact that this might have had on the mix of patients in the ICU and their case-fatality rates (the dependent variable for the severity scores). 

We will leave it to the Editor to adjudicate. If the Editor were unpersuaded by our arguments, we would be happy for the abstract which currently reads as

The study was performed at an Australian tertiary-referral hospital between January 2014 and June 2017, and enrolled consecutive Indigenous and non-Indigenous adults admitted to ICU with sepsis.

To be replaced with

The study was performed at Cairns Hospital, in tropical Australia, between January 2014 and June 2017, and enrolled consecutive Indigenous and non-Indigenous adults admitted to ICU with sepsis.

---

## [Editor Report · Decision Letter 2]

7 Jul 2020

The applicability of commonly used predictive scoring systems in Indigenous Australians with sepsis: an observational study

PONE-D-20-14432R2

Dear Dr. Hanson,

We’re pleased to inform you that your manuscript has been judged scientifically suitable for publication and will be formally accepted for publication once it meets all outstanding technical requirements.

Kind regards,

Biswadev Mitra, MBBS, MHSM, PhD, FACEM

Academic Editor

PLOS ONE
---

## [Editor Report · Acceptance letter]

9 Jul 2020

PONE-D-20-14432R2 

The applicability of commonly used predictive scoring systems in Indigenous Australians with sepsis: an observational study 

Dear Dr. Hanson:

I'm pleased to inform you that your manuscript has been deemed suitable for publication in PLOS ONE. Congratulations! Your manuscript is now with our production department. 

Kind regards, 

on behalf of

Prof. Biswadev Mitra 

Academic Editor

PLOS ONE